# Effects of Ultra-Weak Fractal Electromagnetic Signals on *Malassezia furfur*

**DOI:** 10.3390/ijms24044099

**Published:** 2023-02-17

**Authors:** Pierre Madl, Roberto Germano, Alberto Tedeschi, Herbert Lettner

**Affiliations:** 1Department of Biosciences & Medical Biology, University of Salzburg, A-5020 Salzburg, Austria; 2Prototyping Unit, Edge-Institute at ER-System Mechatronics, A-5440 Golling, Austria; 3Edge-Institute Italia at PROMETE Srl, CNR Spin off, I-80125 Napoli, Italy; 4Research & Development Unit, Edge-Institute Italia at WHB, I-20123 Milan, Italy; 5Laboratory of Environmental Biophysics, Department of Chemistry and Physics of Materials, University of Salzburg, A-5020 Salzburg, Austria

**Keywords:** *Malassezia furfur*, HaCaT, FRACTOS, hormesis, fractal ULF/VLF stimulation, QED

## Abstract

*Malassezia* spp. are dimorphic, lipophilic fungi that are part of the normal human cutaneous commensal microbiome. However, under adverse conditions, these fungi can be involved in various cutaneous diseases. In this study, we analysed the effect of ultra-weak fractal electromagnetic (uwf-EMF) field exposure (12.6 nT covering 0.5 to 20 kHz) on the growth dynamics and invasiveness of *M. furfur*. The ability to modulate inflammation and innate immunity in normal human keratinocytes was also investigated. Using a microbiological assay, it was possible to demonstrate that, under the influence of uwf-EMF, the invasiveness of *M. furfur* was drastically reduced (d = 2.456, *p* < 0.001), while at the same time, its growth dynamic after 72 h having been in contact with HaCaT cells both without (d = 0.211, *p* = 0.390) and with (d = 0.118, *p* = 0.438) uwf-EM exposure, were hardly affected. Real-time PCR analysis demonstrated that a uwf-EMF exposure is able to modulate human-β-defensin-2 (hBD-2) in treated keratinocytes and at the same time reduce the expression of proinflammatory cytokines in human keratinocytes. The findings suggest that the underlying principle of action is hormetic in nature and that this method might be an adjunctive therapeutic tool to modulate the inflammatory properties of *Malassezia* in related cutaneous diseases. The underlying principle of action becomes understandable by means of quantum electrodynamics (QED). Given that living systems consist mainly of water and within the framework of QED, this water, as a biphasic system, provides the basis for electromagnetic coupling. The oscillatory properties of water dipoles modulated by weak electromagnetic stimuli not only affect biochemical processes, but also pave the way for a more general understanding of the observed nonthermal effects in biota.

## 1. Introduction

Yeasts are single-celled eukaryotic microorganisms. Taxonomically, *Malassezia furfur* is grouped under the subkingdom Dikarya, division Basidomycota, order Malasseziales [1]. The genus *Malassezia* comprises lipophilic fungal species whose natural habitat is the skin of humans and other warm-blooded animals [2]. However, *M. furfur* is not only commensal, but it can be involved in the pathogenesis of various dermatological diseases such as folliculitis, *Pityriasis versicolor*, dandruff, *Seborrheic dermatitis*, atopic dermatitis, and psoriasis in individuals with a genetic predisposition [3,4].

The skin, as a physical barrier, is impermeable to most pathogens and thus is the first line of defense. At the same time, it acts also as a chemical barrier by providing cytokines/chemokines, proteases, and antimicrobial peptides/proteins. Keratinocytes secrete a number of soluble factors that are capable of modulating an immune response [5]. Interleukin-(IL)-8, a potent chemoattractant for neutrophils, has been shown to be expressed from keratinocytes in psoriasis and via the stimulation of other cytokines. IL-1β, IL-6, and tumor necrosis factor α (TNF-α) are also important chemical mediators, which are secreted by activated keratinocytes [6] in the acute inflammatory phase and during the healing process of skin lesions alike [7]. Hence, *Malassezia* species may act as controllers of innate immunity. Ishibashi et al. [8] found that *M. globosa* induced the expression of IL-6 in normal human epidermal keratinocytes (NHEKs), while Baroni et al. [9] showed that *M. furfur* upregulated the gene expression of IL-8 (along with toll-like receptor (TLR-2) and human-β-defensin-2 (hBD-2) when co-cultured with *M. furfur*.

Antimicrobial peptides of the hBD family are expressed in all human epithelial tissues as they represent an ancient branch of the innate immune system whose role is to promptly neutralize invading microbes [10]. Conventional therapeutic countermeasures for the treatment of diseases, such as *Malassezia folliculitis*, *Pityriasis versicolor*, dandruff, and *Seborrheic dermatitis,* rely on azoles (antifungal agents such as ketoconazole, itraconazole, fluconazole, etc.), which have an inhibitory effect against *Malassezia* species. Skin diseases associated with *Malassezia* species are chronic and recurrent disorders, requiring repeated therapy. Photodynamic inactivation (PDI), for example, is a novel and effective physicochemical therapeutic option [11]. It relies on two components—electromagnetic radiation in the visible range combined with a photosensitizer as an additive—to unfold the antimicrobial effect. While PDI operates already with low-dose photosensitiser concentrations, bio-electromagnetism can operate with even lower dosages, revealing a biphasic dose–response relationship [12], i.e., a low dose stimulation manifests itself as a beneficial effect, whereas it results in inhibitory effects when administered in high doses. Although ample literature has accumulated over past decades on the adverse effects of high-dose electromagnetic stimulation (see selected publications [13,14,15,16,17,18,19,20,21]), there are currently only a few studies available addressing the low-dose effect of electromagnetic stimulation ([22], and references therein)—for details see the Appendix B. Given that hormetic effects per se are already difficult to perceive under the current biological paradigm, we feel there is a need to bridge the prevailing atomistic/molecular worldview with the field aspect initiated by Schrodinger [23]. Over the decades that followed, this branch of physics elaborated a respectable body of knowledge that no longer restricts itself to solid-state physics, but gradually makes its way into soft-matter physics, with quantum electrodynamics (QED) being the foremost promoter—for details we likewise ask the reader to consult the Appendix B.

In this investigation, we utilize ultra-weak fractal electromagnetic (uwf-EM) signals that have already been used to stimulate biological samples [24,25,26]. This concept has recently been further refined using a revised prototype termed FRACTOS [26]. However, in doing so, we used an updated version of this prototype to investigate the response of HaCaT cells infected with *Malassezia furfur* by treating both with a uwf-EM signal. This device essentially consists of a logarithmic coil printed on a circuit board, connected to a signal source, which provides an audio signal (in the 0.5 to 20 kHz range) and whose loudness–frequency relation has fractal properties [27]. The logarithmic coil progression was chosen as it occurs in all domains of life and is expressed in the physiological and functional properties of living systems [28]. Thus, we demonstrate with this approach that such signals can indeed modulate the inflammatory response of infected HaCaT cells and enhance cytoprotective responses via hBD-2.

## 2. Results

### 2.1. Effect of uwf-EMF Treatment on Malassezia furfur Invasivity

Yeast cells plated on Sabouraud agar and treated with the uwf-EM signal showed a reduction in growth dynamics compared to untreated cells. As shown in Figure 1 (respectively, Table 1), a 35% reduction in *M. furfur* invasiveness was found after 72 h with respect to *M. furfur* batches for both HaCaT-exposed (effect size d = 2.456 with a significance value *p* < 0.001) as well as HaCaT-unexposed cells (effect size d = 3.555 with a significance value *p* = 0.018). To determine whether *M. furfur* is capable of affecting the invasiveness of the yeast controls, HaCaT cultures were infected with uwf-EM pre- and untreated *M. furfur* cultures (at a 1:30 cell:yeast ratio). After 48 h, the effect of uwf-EM exposure on *M. furfur*‘s invasiveness was assessed.

### 2.2. Effect of uwf-EM Signals on HaCaT Cells Exposed to Malassezia furfur

Human keratinocytes were cultured with or without *M. furfur* in a 30:1 ratio (yeast cells to keratinocytes). To monitor the cell proliferation of HaCaT cells, spectrometric absorbance measurements were performed every 24 h - shown in Figure 2 (respectively, Table 2). The corresponding optical density corresponds to the mass concentration of the HaCaT cells infected with or without *M. furfur* as well as with and without uwf-EM treatment. The steadily increasing absorbance with time (accumulation of formazan concentration within and on the cell surfaces) indicates that the mitochondrial reductase of the HaCaT cells maintained adequate vitality, even after 72 h of incubation. This implicates that uwf-EM treatment did not alter the viability of the cells.

### 2.3. Effect of uwf-EM Signals on the Proinflammatory Response of HaCaT Cells Infected with Malassezia furfur

To investigate the potential of uwf-EM exposure to modulate the inflammatory response, we also evaluated gene the expression of cytokines in noninfected HaCaT cells as well as in infected ones with and without uwf-EM pre-treated *Malassezia*. As shown in Figure 3 (respectively, Table 3), we detected a marked increase in the IL-6, IL-8, and IL-1α gene expression in the HaCaT cells following an infection with *M. furfur* after 24 h of incubation; the infection boosted IL-6 and IL-1α by a factor of ≈3 compared to uninfected HaCaT cells, while IL-8 increased almost by a factor of ≈5.

In contrast, in HaCaT infected batches with pre-treated *M. furfur* that underwent uwf-EM exposure, we observed that the IL-6 and IL-8 gene expressions fell again by 37.5% and 44.2%, respectively, when compared to HaCaT cultures infected with *M. furfur* not exposed to the uwf-EM field. IL-1α concentrations decreased to a lesser extent by 21.5%.

### 2.4. Effect of uwf-EM Signals on hBD-2

To investigate the expression of hBD-2, keratinocytes were infected for 24 h with and without uwf-EM pre-treated *M. furfur*. Following the 24 h incubation period, as shown in Figure 4 (respectively, Table 4), the batch infected with *M. furfur* that did not undergo uwf-EM exposure resulted almost in double the hBD-2 compared to the HaCaT controls. The batch infected with *pre*-treated *M. furfur* that did undergo uwf-EM treatment almost tripled with respect to the HaCaT infected with *M. furfur* only.

## 3. Discussion

Keratinocytes are the major constituent of the outermost epidermal layer and, as such, not only provide the keratin that gives human skin its strength, but also shield organisms from environmental pathogens. Under normal circumstances, the commensal fungus *Malassezia* spp. is part of a healthy cutaneous microbiome that colonizes the skin without causing harm. Yet in immunosuppressed individuals, the proliferation of this saprophyte can result in a wide range of skin-related disorders. Therefore, immortalised human keratinocytes (HaCaT) have been used to study the interaction of *M. furfur* in the presence of a low-intensity fractal electromagnetic signal in the ultra-low (ULF) to very-low frequency (VLF) bands. As both the fractal property and the low intensity of the applied field seem to contradict common sense and given the fact that the concept of hormesis still remains a challenge to the biomedical and clinical communities [29], a detailed look at these two aspects is given in the Appendix B. Therein, we provide insight for a better understanding of the therapeutic implications of the hormetic dose–response relationships. In addition, we attempt to bridge the gap between hormesis and coherence in living matter, as was already addressed to some extent in our previous study [27].

Keratinocytes release cytokines after microbial stimulation, and *M. furfur* has repeatedly been shown to modulates the production of proinflammatory mediators in keratinocytes [30]. In the case of skin infection or injury, the expression of antimicrobial peptides is upregulated. Since the gene expression of hBD-2 is virtually absent in healthy skin, the expression in HaCaT cells can only be stimulated via the exposure of cutaneous pathogens or be induced by other environmental stressors [31].

In this study, we evaluated the ability of uwf-EM signals to modulate the hBD-2 gene expression in HaCaT cells. Our results showed that uwf-EM exposure of both *M. furfur* and infected HaCaT cells is able to increase the hBD-2 gene expression in keratinocytes by a factor of ≈3 compared to the nonexposed controls. This is remarkable since defensins attract inflammatory cells, such as neutrophils, B-cells, and macrophages [32,33]. Furthermore, our experiments demonstrate that HaCaT cells respond with a strong increase in IL-6 and IL-8 gene expressions after 24 h following infection with *M. furfur* compared to untreated cells. In contrast, uwf-EM-treated *M. furfur* and infected HaCaT cells resulted in suppressed IL-6 and IL-8 gene expressions when compared to the *M. furfur*-infected HaCaT culture. In parallel, we examined the invasive capability of uwf-EM-treated *M. furfur* and demonstrated that the invasiveness of *M. furfur* was subsequently reduced by 35% compared to the non-uwf-EM-exposed controls. This finding suggests that treatment with uwf-EM signals has the potential to reduce the incidence of systemic mycosis.

Our results also suggest that the huge upregulation of hBD-2 following exposure to the uwf-EM field represents one of the primary innate immune responses against microorganisms such as *M. furfur*. As expected, proinflammatory cytokines, such as IL-6, IL-8, and IL-1α, as well as the antimicrobial peptide hBD-2 are upregulated upon infection with *M. furfur*. Exposure with uwf-EM treatment, however, supresses the gene expression of cytokines, while the expression of the antimicrobial peptide is further stimulated.

In terms of the potential medical applications, keratinocytes are an ideal model for analysing the biological effects of ultra-weak (low-intensity) nonionizing ULFs/VLFs. Elucidating a deeper understanding of the mechanisms could lead to novel nonchemically based methods against *M. furfur* infection. From a therapeutic perspective, uwf-EM exposure could be used as a basic treatment in conjunction with conventional therapy when treating *M. furfur*. As interesting as these results may be, further in vivo studies are necessary to understand the physicochemical processes triggered by these fields. Promising examples utilizing similar approaches are currently in the making and concern the use of pulsed magnetic field treatment for diabetic cells [34] and narrow-band electromagnetic field exposure in containing COVID-19 [35] as well as altering the proton density and ATP synthase rotation in the context of mitochondrial dysfunction, as often observed in tumour biology [36]. In fact, tumour-treating fields are already used in some medical fields. These rely on the application of a 1–3 V/cm (0.335–1.005 µT) field in the β-dispersion regime (100–300 kHz) to induce a regression in glioblastomas by adversely affecting the microtubule assembly during mitotic events [37].

## 4. Materials and Methods

### 4.1. Microorganisms and Culture Media

*Malassezia furfur* ATCC 12078 was obtained from the American Type Culture Collection (Rockville, MD, USA). *M. furfur* was grown for 4 days at 30 °C in Sabouraud’s dextrose agar containing peptone (1%), glucose (4%), olive oil (2.0%), and Tween 80 (0.2%).

### 4.2. Cell Culture and Treatments

Aneuploid immortal keratinocytes (HaCaT) from adult human skin were plated in 6-multiwell plates (35 mm diameter) with 2 mL of Dulbecco’s modified Eagle medium (DMEM) supplemented with 10% foetal bovine serum, 100 U/mL penicillin, 100 μg/mL streptomycin, and 2 mM L-glutamine at 37 °C and 5% CO_2_.

*M. furfur* was washed three times in phosphate-buffered saline (PBS). The resulting precipitate was resuspended in a small volume of Dulbecco’s modified Eagle medium (DMEM) and vortexed gently to avoid yeast aggregation. Thereafter, one batch of *M. furfur* designated for the exposure trials was treated for 10 min under dark conditions with the uwf-EM-generating device at a working distance of 8 cm from the culture. The control batch was handled in the same way but was sham-exposed (i.e., the uwf-EM device was disabled). The exposure procedure was conducted on a tray-like setup where the antenna was placed atop, while the biological samples were placed below using a working distance of 80 mm. Since proper field propagation is essential, it was of paramount importance that no ferromagnetic material was located nearby. To ensure a standardized uwf-EM field exposure, a fully digitized prototype was used. This field-generating device was based on the WHITE holographic bioresonance method [24] and consists of a logarithmic coil printed on a circuit board. This “antenna” is supplied with a VLF/ULF audio signal which has fractal characteristics. Only a fully digitized approach allows for standardized field exposures. In order to achieve this, a computer-based solution (Rasbperry-Pi^®^) was chosen, which allows the user to select the proper fractal audio file via menu-guided options. Thus, the signal-generating source consists of three sub-units: (i) the processor board, (ii) the touchscreen, and (iii) the high-end D/A converter. Together with the PCB antenna, it forms a unit we refer to as FRACTOS (for more details see [27]). Incubation following uwf-EM treatment was executed in separate compartments of the incubator to avoid electromagnetic crosstalk, which could induce artifacts [38].

Yeast viability was assessed using standard colony-forming unit (CFU) counts. Human keratinocytes were treated with *M. furfur* at a ratio of 30:1, were likewise treated with the uwf-EMF signal using the same device (except the controls), and were incubated for up to 96 h. A simplified process flow diagram (not showing the uwf-EM treatment of the *M. furfur* culture) can be seen in Figure 5. Since cell proliferation rate of the keratinocytes was similar to that of the uninfected controls, it can be assumed that the vitality of the keratinocytes is not affected by *M. furfur*. After 24 h and 48 h, the keratinocytes were processed for RNA extraction. Morphological features of HaCaT cells were defined by phase-contrast microscopy using an Olympus CDK40 at 20× magnification (not shown).

### 4.3. MTT Cell Proliferation Assay

The metabolic activity was measured using an MTT assay. HaCaT cells were grown in microplates (tissue culture grade, 96 wells, flat bottom) in a final volume of 100 μL of DMEM at 37 °C and 5% CO_2_. After 24 h of treatment, 10 μL of 3-(4,5-dimethylthiazol-2-yl)-2,5-diphenyltetrazoliumbromide (MTT) labelling reagent (Roche Diagnostics, Basel, Switzerland; final concentration 0.5 mg/mL) was added to each well. Following the procedure according to [39], 100 μL of the solubilization solution (10% SDS in 0.01 M HCl) was added to the culture after 4 h and was incubated overnight. The spectrophotometric absorbance was measured using a microplate ELISA reader (Biorad) at a 570 nm wavelength as the conversion from yellow tetrazole to purple formazan by the living cells is best seen at this bandwidth.

### 4.4. Microorganisms and Culture Media

To distinguish between uwf-EM-treated and untreated *M. furfur*, two parallel batches of semi-confluent keratinocytes (106/well) were needed and infected with *M. furfur*. After 24 h of incubation, the total RNA content was isolated using the High Pure RNA Isolation Kit (Roche; Milan, Italy). Approximately 200 ng of the total cellular RNA was reverse transcribed (Expand Reverse Transcriptase, Roche; Milan, Italy). Transcription into complementary DNA (cDNA) using random hexamer primers (Random hexamers, likewise from Roche) occurred at 42 °C for 45 min. Real-time PCR was carried out with the LC Fast Start DNA Master SYBR Green kit (Light Cycler 2.0 Instrument, also from Roche) using 2 mL of cDNA, corresponding to 10 ng of total RNA, in a final volume of 20 mL to which 3 mM MgCl_2_ and 0.5 mM sense and antisense primers were added (Table 5). A melting curve was produced at the end of each amplification to ensure that no nonspecific reaction products were present. As the accuracy of mRNA quantification depends on the linearity and efficiency of the PCR amplification, both parameters were assessed using standard curves generated by increasing amounts of cDNA. Quantification was based on the threshold cycle values measured at the early stage of the exponential phase of the reaction. Normalization to the internal standard curve with the β-actin gene was also performed to avoid discrepancies in the input RNA or reverse transcription efficiency. Finally, PCR products were examined in a 1.4% agarose gel electrophoresis.

### 4.5. Invasion Assay for Malassezia furfur

Human keratinocytes were infected with and without uwf-EM-treated *M. furfur*. Infection occurred at a dilution rate of 30:1 (yeast-to-keratinocyte ratio, as proposed by Donnarumma et al. [40]) for 24 and 48 h. To remove nonadherent yeast cells, the keratinocytes were washed three times with PBS. The infected keratinocytes were successively treated with 1 mL of DMEM containing ketoconazole at the micocidal concentration and were incubated for 4 h at 37 °C. The infected cells were then treated with trypsin-EDTA for 5 min at 37 °C and lysed by adding 1 mL of cold 0.1% Triton-X100. The cell lysates were diluted in PBS, plated on Sabouraud–dextrose agar, and incubated for 72 h at 30 °C. The viable intracellular yeast content was then determined and plotted as a concentration mL^−1^.

### 4.6. Statistical Analysis

Both experiments and measurements were performed in triplicate, enabling basic statistics. Beginning with the magnetic flux density measurements, the results are expressed in Table 6 as mean ± standard deviations (±SD). The measurements were taken using an NFA-1000 magnetometer with an integrated data logger (GHz-Solutions, FRG) and were processed with the software packages (NFAsoft v.1.72 from the same supplier). Due to the weak signals irradiated from the antenna, the measurements were performed inside a mu-metal test chamber (Marchandise-Tech, FRG) specifically designed for this type of investigation. For all exposure trials, the following uncompressed source file was used: S1_20-24_bit 48k_with_cutoff_500 Hz.wav.

To estimate the effect size of the CFUs of *M. furfur*, Cohen’s “d” was determined. According to Cohen’s conventions, there is a significant difference in the means if d > 1. For both the uwf-EM-exposed and *M. furfur* in contact with HaCaT cells, values of 3.56 (uwf-EM “on/off”) and 2.46 (uwf-EM “on/off” when in contact with HaCaT cells) standard deviations were obtained.

Assuming that the repetitive trials belong to a Gaussian distribution, the probability was determined using a *t*-test and the corresponding *p*-value. For this purpose, a standard significance level alpha (α) of 0.05 was defined. For all the graphs in the figures above, it can be shown that the probability that a random gene expression could have provoked the differences is extremely low and that the results obtained are indeed significant differences with regard to the controls.

To determine the statistical significance of the PCR gene expression results an ANOVA test (analysis of variance between groups) was used (Table 7). The hypothesized mean difference (with H_0_ stating that there is no difference between means) was set to zero. As with the above case, a standard alpha level (α) 0.05, or 5%, was chosen. The *p*-value for the proinflammatory response of the yeast-infected HaCaT cells was found to be <0.002, confirming the statistical significance of the results. It was also found that the antimicrobial and cytotoxic peptide in the form of hBD-2 fluctuated to some extent in the controls, whereas the yeast-infected and uwf-EM-treated keratinocytes showed a somewhat stabilized defensin production. The documented differences are significant, suggesting that yeast infection and uwf-EM treatment do indeed upregulate the hBD-2 expression.

## 5. Conclusions

*Malassezia furfur* is a yeast fungus that is an integral part of an intact skin microbiome (present in more than 90% of adult humans). However, in stressed individuals and due to environmental factors, it can occur as an opportunistic pathogen. Therefore, its presence is considered important in the etiology of skin disorders such as *Pityriasis versicolor* and *Seborrheic dermatitis*. *M. furfur* was collected from adult individuals by swabbing and was transferred to Sabouraud agar, placed in saline solution, and exposed for 10 min to an uwf-EM field.

The pre-treated yeast (for the exposure trials) and the untreated yeast (for the controls) were used to infect HaCaT cells (immortalised human keratinocytes) at a ratio of 30:1. The infected cells were incubated for up to 96 h, were repeatedly uwf-EM-exposed, and were repeatedly processed for RNA extraction to assess their gene expression for markers normally involved in the inflammatory process as a result of *M. furfur* infection. In this study, the application of uwf-EM stimuli was shown to have regulatory effects on gene expression, which was reflected in the reduced invasiveness of *M. furfur* without drastically affecting the growth dynamics in HaCaT cells. Considering the very low intensities of the applied uwf-EM signals, we can clearly label the EM-induced effect as hormetic. The low-intensity stimulation induced cellular adaptations, as previously observed by Vaiserman [41], and stimulated protein activation in the living system, documented by Kim et al. [42]. Although the uwf-EM exposure time between incubation cycles was quite short (10 min each), it was apparently sufficient to induce a genetic adaptation response. This led to a long-lasting memory effect (probably of epigenetic origin) that needs to be investigated in future studies. In turn, it renders adaptive responses more efficiently and corroborates a positive hormetic feedback principle. The use of uwf-EM agents (hormetins) offers a fascinating area for further research, with reprogramming of the epigenome to reestablish a homeostatic healthy steady state. This has the potential to become the method of choice as the undesired side effects so often observed with biochemical agents can be minimized.

## Figures and Tables

**Figure 1 ijms-24-04099-f001:**
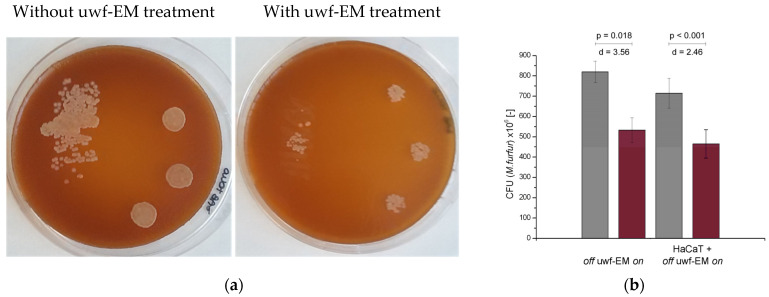
CFUs conducted on *M. furfur* in three replicates after 48 h of incubation (**a**) with (**right pane**) and without (**left pane**) uwf-EM pre-treatment. (**b**) Quantitative representation of CFU-counts of *M. furfur.* Colour codes: gray, without uwf-EM stimulation, red with uwf-EM stimulation. Both the calculated effect sizes (d) and *p*-values (*p* < 0.05) underline the statistical significance of the results.

**Figure 2 ijms-24-04099-f002:**
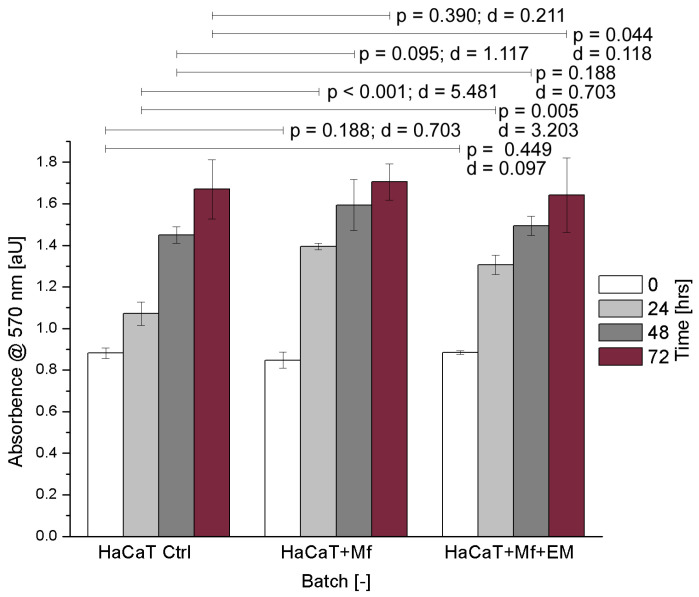
HaCaT cell cultures showing the control, *M. furfur*-infected, and *M. furfur*-infected cells that have been treated with the uwf-EM signal. Both the calculated effect sizes (d) and *p*-values (*p* < 0.05) highlight the inexistence of a statistically significant difference after 72 h with *p* = 0.390 vs. d = 0.211 for HaCaT + *M. furfur* exposure and *p* = 0.044 vs. d = 0.118 for HaCaT + *M. furfur* + uwf-EM exposure combined.

**Figure 3 ijms-24-04099-f003:**
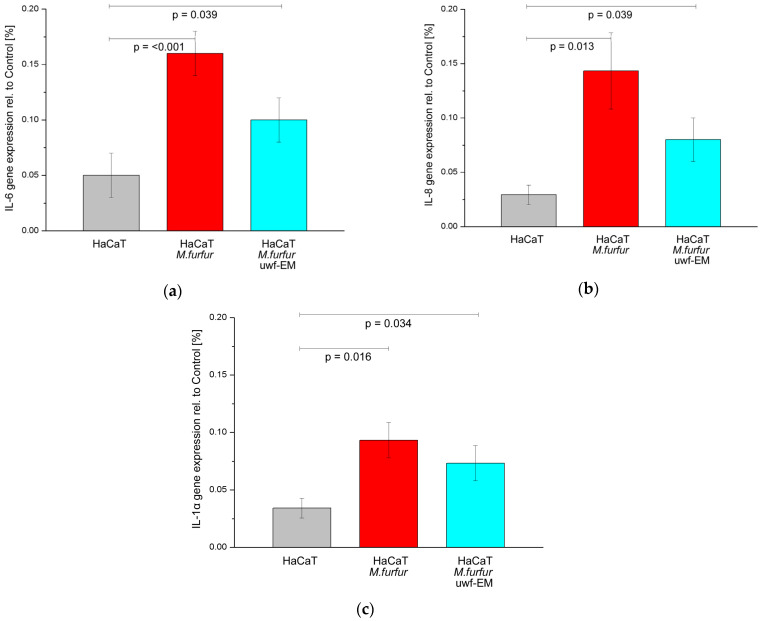
Gene expression of IL-6 (**a**), IL-8 (**b**), and IL-1α (**c**) of HaCaT cell cultures after 24 h of incubation. The figure depicts the control (grey), the batch infected with *M. furfur* (red), and the batch infected with pre-treated *M. furfur* that underwent uwf-EM exposure (blue). For further details, see the Methods section. The calculated *p*-values (*p* < 0.05) highlight the statistical significance of the obtained results.

**Figure 4 ijms-24-04099-f004:**
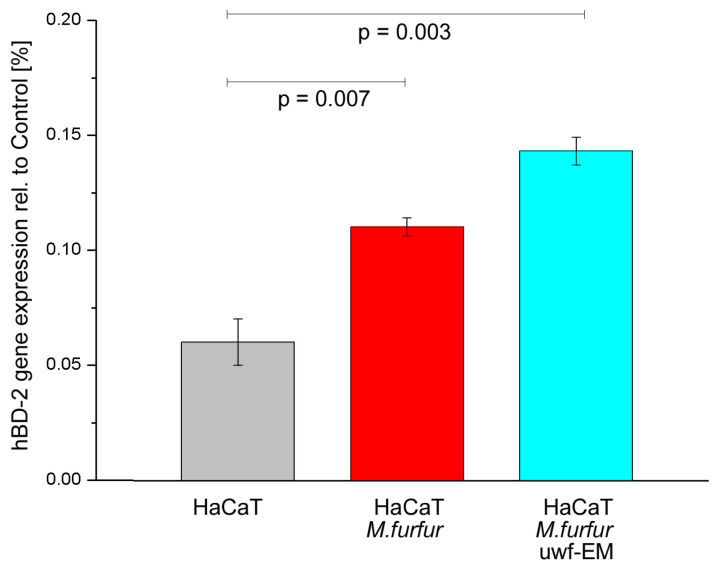
hBD-2 gene expression of HaCaT cell cultures after 24 h of incubation. The figure shows the control (grey), the batch infected with *M. furfur* (red), and the batch infected with pre-treated *M. furfur* that underwent uwf-EM exposure (blue). For further details, see the Methods section. The calculated *p*-values (*p* < 0.05) highlight the statistical significance of the obtained results.

**Figure 5 ijms-24-04099-f005:**
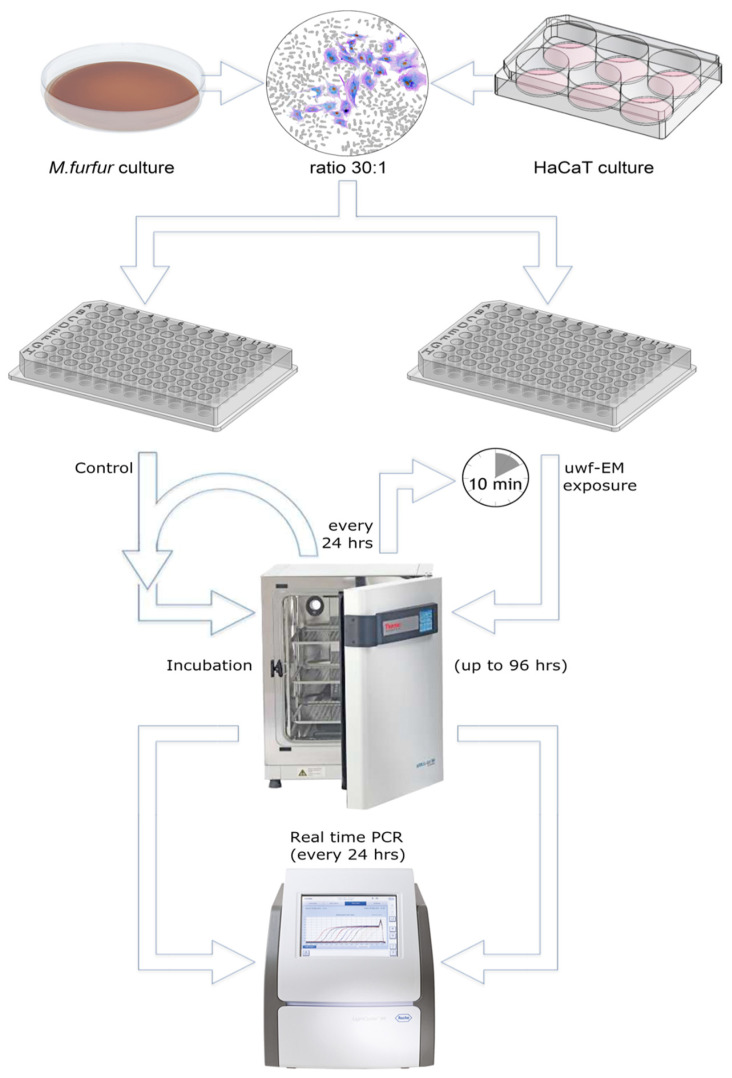
Simplified sketch of the performed protocol. HaCaT cells were cultivated in 6-well plates, whereas *M. furfur* was cultivated on Sabouraud agar in a 100 mm Petri dish. Initially, fungal cultures were uwf-EM pre-treated for 10 min (at a distance of approx. 80 mm from the log antenna), while controls were not treated (not shown in the figure). Infection of keratinocytes with the yeast was performed in 6-well plates (35 mm) at a ratio of about 30:1. Thereafter, incubation of infected and uninfected cell cultures for up to 96 h started. HaCaT viability was assessed by an MTT assay, while the gene expressions of IL-6, IL-8, and IL-1α were performed using RT-PCR. Every 24 h, the charge to be tested was uwf-EM-exposed for 10 min followed by another cycle of incubation with the same distance parameters mentioned above.

**Table 1 ijms-24-04099-t001:** CFU counts of *M. furfur* following 48 h of incubation. Both emf-EM-exposed cultures (those in contact with and without HaCaT cells) show a 35% reduction in CFUs compared to non-uwf-EM-exposed controls.

48 h (CFU × 10^6^)	Trial 1	Trial 2	Trial 3	Average ± STDev
uwf-EM off	832	762	866	820 ± 53.0
uwf-EM on	466	584	550	533 ± 60.7
HaCaT + uwf-EM off	789	714	641	715 ± 74.0
HaCaT + uwf-EM on	534	467	395	465 ± 69.5

**Table 2 ijms-24-04099-t002:** Numeric representation of CFUs versus time as averages of three replicates along with the corresponding standard deviations.

Time [h]	HaCaT Ctrl	HaCaT + *M. furfur*	HaCaT + *M. furfur* + uwf-EM
0	0.881 ± 0.026	0.848 ± 0.039	0.884 ± 0.009
24	1.071 ± 0.057	1.395 ± 0.016	1.307 ± 0.047
48	1.450 ± 0.040	1.595 ± 0.123	1.494 ± 0.047
72	1.670 ± 0.143	1.705 ± 0.088	1.643 ± 0.179

**Table 3 ijms-24-04099-t003:** Numeric representation of pro-inflammatory cytokines compared to the control as averages of three replicates along with the corresponding standard deviations.

Gene	HaCaT	HaCaT + *M. furfur*	HaCaT + *M. furfur* + uwf-EM
IL-6	0.047 ± 0.014	0.160 ± 0.016	0.100 ± 0.016
IL-8	0.029 ± 0.007	0.143 ± 0.028	0.080 ± 0.016
IL-1α	0.034 ± 0.007	0.093 ± 0.012	0.073 ± 0.013

**Table 4 ijms-24-04099-t004:** Numeric representation of the antimicrobial peptide compared to the control as averages of three replicates along with the corresponding standard deviations.

Gene	HaCaT	HaCaT + *M. furfur*	HaCaT + *M. furfur* + uwf-EM
hBD-2	0.060 ± 0.005	0.110 ± 0.004	0.143 ± 0.006

**Table 5 ijms-24-04099-t005:** Sense and antisense primers of the five gene sequences showing sense (upper row) and antisense strands (lower row).

Gene	Primer Sequence	Conditions	Product Size (bp)
IL-6	5′-CTC CAG CAT CCG ACA AGA AGC-3′ 5′-GAG GTC GTA GGC TGT TCT TCG-3′	1′ at 94 °C, 1′ at 63 °C, 1′ at 72 °C for 33 cycles	234
IL-8	5′-ATG ACT TTC AAG CTG GCC GTG-3′ 5′-TGA ATT CTC AGC CCT CTT CAA AAA CTT CTC-3′	1′ at 94 °C, 1′ at 56 °C, 1′ at 72 °C for 33 cycles	297
hBD-2	5′-CCA GCC ATC AGC CAT GAG GGT -3′ 5′-AAC CGG TAG TCG GTA CTC CCA-3′	1′ at 94 °C, 1′ at 63 °C, 1′ at 72 °C for 33 cycles	254
IL-1α	5′-CCG ACT ACT ACG CCA AGG AGG TCA CGT-3′ 5′-AGG CCG GTT CAT GCC ATG AAT GGT GCA-3′	1′ at 94 °C, 1′ at 60 °C, 2′ at 72 °C for 32 cycles	439
β-actin	5′-TGA CGG GGT CAC CCA CAC TGT GCC CAT CTA-3′ 5′-CTA GAA GCA TTG CGG GTG GAC GAT GGA GGG-3′	30″ at 95 °C, 1″ at 56 °C, 30″ at 72 °C for 35 cycles	661

**Table 6 ijms-24-04099-t006:** Magnetic flux densities of the background with the antenna in operation stated as averages along with standard deviations of three replicates. Background values denote the signal strength with both the inactivated antenna and cell culture placed in the mu-metal test box, whereas uwf-EM measurements denote the emitted field intensities when the antenna was operational.

Trials	Min	Max	Avg	Stdev	95th%ile	Unit
Backgnd.	0.500 ± 0.001	3.033 ± 0.808	0.967 ± 0.023	0.377 ± 0.040	1.680 ± 0.125	[nT]
uwf-EM	7.133 ± 0.379	12.567 ± 0.208	9.310 ± 0.141	0.580 ± 0.040	10.260 ± 0.185	[nT]

**Table 7 ijms-24-04099-t007:** Single-factor ANOVA test. With reference to the figures above, the divergent trends of the HaCaT controls, the cultures infected with *M. furfur,* and the cultures also exposed to uwf-EMF yield significant differences at a significance level of *p* < α (0.05). Accordingly, H_0_ was rejected for IL-6, IL-1α, IL-8, and hBD-2, as confirmed by the corresponding F-values, which are much greater than those of F_crit_ (>5.143).

ANOVA	F	*p*	F_crit_
IL-6	22.750	0.002	5.143
IL1-α	14.978	0.005	5.143
IL-8	17.124	0.003	5.143
hBD-2	119.647	<0.001	5.143

## Data Availability

The data presented in this study are available in Appendix A.

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
