# Peer review of "Effects of Ultra-Weak Fractal Electromagnetic Signals on *Malassezia furfur"

_ijms, 2023, doi:10.3390/ijms24044099_

Round 1

Reviewer 1 Report

This is a fascinating paper which may be very important. I found it very difficult to interpret the figures for example in figure 3 no measure of the level of significance is given and the legend does not allow independent understanding of the result shown. In figure 4 the same applies. The description of the results is also confusing CFU implies that actual colonies were counted but it appears no clonogenic assays were done and the results are based on a cell viability assay. The other issue is the use of appendices which cover over 75% of the cited references. While I appreciate the need to "convert" people to accept hormesis, I think a condensed description of the process should be in the introduction with appropriate references. Ditto for the description of the ultraweak EMF description. English needs improvement throughout.

Author Response

Reply from the authors:

We thank the reviewer for her/his detailed comments and suggestions.  In a first step, we rewrote the manuscript to improve the English used. Doing so, we hope that the manuscript in its current form meets the linguistic requirements of the reviewer.

Regarding the possibilities for improvement according to the selection table, we would like to answer as follows:

  1. i) Does the introduction provide sufficient background and include all relevant references? During our post-processing work, we tried to restructure the introduction and make it more readable. We also added a paragraph about Hormesis (line 62 ff):

While PDI operates already with low dose photosensitizer concentrations, bioelectromagnetism can operate with even lower dosages.  Indeed, hormetic stressors reveal a biphasic dose response relationship (Mattson, 2008)[1]; I,e, a low dose stimulation manifests itself as a beneficial effect, whereas produced inhibitory effects when administered in high doses. Although amble literature has accumulated over past decades on the adverse effect of high-dose electromagnetic stimulation (selected publications include: Ayrapetuan & Markov, 2006;[2] Becker & Marino, 1982;[3] Koenig et al., 1981;[4] Marha et al, 1971;[5] Marino, 1989;[6] or Markov, 2015;[7] Presman, 1970;[8] Smith, 1989;[9] Zhadin, 2001[10]) there are only a few studies available addressing the low-dose effect of electromagnetic stimulation - for details see Appendix A.  Given that hermetic effects per se are already difficult to perceive under the current biological paradigm, wee feel there is a need to bridge the prevailing atomistic/molelcular world view with the field aspect initiated by Schroedinger (1944).[11] Over the decades that followed, this branch of physics elaborated a respectable body of knowledge that was no longer restricted to solid state physics, but gradually made its way into soft matter physics, with QED being the foremost promoter. for details see Appendix B

  1. i) Are all the cited references relevant to the research? The source literature we have consulted concerns only those works that are relevant to our study. Of course, this list can be extended, but it will hardly be possible to integrate all the medical studies that would be relevant for this purpose. Given that the references used are quite extensive, we decided to skip part of the theoretical ones used in the appendix – see below.
  2. i) Is the research design appropriate? The research design follows the same pattern as performed in our previous investigation (see Madl et al., 2020).[12] Therefore we hope that the reviewer will accept it as it is. Otherwise, the entire experimental work needs to be redone and given the limited resources available at the moment this may put a complete halt to the research-work we are currently engaged in.
  3. i) Are the methods adequately described? During the revision process of this manuscript, the methodological part, especially the statistics section, has been enhanced.
  4. i) Are the results clearly presented? In the rework process it became clear to us that the presentation of the results needs of improvement. We have therefore upgraded this section and hope it will be compliant with the reviewer.
  5. i) Are the conclusions supported by the results? Given the positive feedback of the reviewer “This is a fascinating paper which may be very important” we hope that the revision of this part of the manuscript likewise contributed to a better and clearer read than was the case with the original version.
  6. i) Comments and Suggestions for Authors:

in fact, the submitted version was difficult to understand.  With the revised version, we hope that we have been able to correct this aspect as well. Having done so we hope that we could convey the implications of our work in a much better way.

Figure 3 and its legend have been improved - it now also includes statistical parameters.

Figure 4 has also been reworked accordingly. Unfortunately, we made a mistake in the submitted version; i.e. instead of embedding the correct dataset in the graphing software package, an erroneous dataset was used that did not correspond to the laboratory results.  We apologize for this and have mapped the correct dataset in the current version.

The presentation of the CFUs of M.furfur in figure 1 has also been corrected and trimmed to the essential results.

Given the rather heavy weightiness of the two appendices, we agree with the reviewer and decided to shorten it.  Practically, we think that the explanations of hormesis and QED given in the appendix (formely appendix A) refers to the scope of the special issue (The Emerging Role of Quantum Sciences and Radiation Biology in Biomedical Applications) and therefor decided to keep it. Appendix B has been skipped and will be used for a more theoretical paper that we are currently working on.

Hoping that the editing done in the revised version of our manuscript will find the approval by the reviewer.

Looking forward to the reviewers reply

Kindest regards

Pierre Madl et al.

[1] Mattson, M.P. Hormesis defined. Ageing Res Rev. 2008; 7(1):1-7. doi: 10.1016/j.arr.2007.08.007.

[2] Ayrapetyan, S.N., Markov, M.S. Bioelectromagnetics Current Concepts. NATO security through science series. Series B, Springer, Dodrecht (NL). 2006. doi 10.1007/1-4020-4278-7

[3] Becker, R.O., Marino, A.A. Electromagnetism and Life. New York State University Press, Albany (NY). 1982. ISBN: 978-0981-8549-08

[4] Koenig, H.L., Krueger, A.P., Lang, S., Sonning, W. Biologic Effects of Environmental Electromagnetism. Springer, Berlin, (FRG). 1981. doi: 10.1007/978-1-4612-5859-9

[5] Marha, K., Musil, J., Tuha, H. Electromagnetic Fields and the Life Environment. San Francisco Press (CA). 1971. ISBN: 91-1302-13-7

[6] Marino, A.A. Modern Bioelectricity, CRC Press, Boca Raton (FL). 1988. doi: 10.1201/9781003065821

[7] Markov, M.S. Electromagnetic Fields in Biology and Medicine. CRC Press. Boca Raton (FL). 2015. doi: 10.1201/b18148

[8] Presman, A.S. Electromagnetic Fields and Life. Springer, New York (NY). 1970. doi: 10.1007/978-1-4757-0635-2

[9]  Smith CW, Best S (1989) Electromagnetic Man – Health and Hazard in the Electrical Environment. St. Martin’s Press, (NY), USA. ISBN: 978-0312-0373-07

[10] Zhadin MN. Review of russian literature on biological action of DC and low-frequency AC magnetic fields. Bioelectromagnetics. 2001; 22(1):27-45. doi: 10.1002/1521-186x(200101)22:1<27::aid-bem4>3.0.co;2-2.

[11] Schroedinger, E. What Is Life? The Physical - Aspect of the Living Cell. 13th ed. Cambridge University Press. 1944. ISBN: 978-1-107-60466-7

[12] Madl. P.; De Filippis, A.; Tedeschi, A. Effects of ultra-weak fractal electromagnetic signals on the aqueous phase in living systems: a test-case analysis of molecular rejuvenation markers in fibroblasts. Electromagn. Biol. Med., 2020, 39(3): 227-238.  doi: 10.1080/15368378.2020.1762634

Reviewer 2 Report

The paper describes the effect of ultra-weak fractal electromagnetic (uwf-EMF) signals on the growth dynamics and invasiveness of M. furfur and the effect of “magnetized M. Furfur on cytokine expression in skin derived cell lines. The paper could be strengthen with addressing the points below: 

Major:

1.     All the graphs are missing statistical analysis and number of repetitions 

2.     Appendix A should be excluded. It is not based on any experimental results, and it is not too scientific. If want to include, experimental data should be presented 

3.     It would be good to show results from HaCat cells + EM 

Minor:

1.     The figures are not in the right order. Starts from figure 2. 

2.     Please provide more details early in the text the electromagnetic set up and what is the field strength in mTesla of the EM. 

3.     Line 78 should be mentioned that the reduction was observed only after 48 hr and not 24 hours. 

4.     4.2 still contain traces of the track changes in red and blue

Author Response

We thank the reviewer for her/his detailed comments and suggestions.  In a first step, we rewrote the manuscript to improve the English used. Doing so, we hope that the manuscript in its current form makes it easier to read.

Regarding the possibilities for improvement according to the selection table, we would like to answer as follows:

  1. i) Are all the cited references relevant to the research? The source literature we have consulted concerns only those works that are relevant to our study. Of course, this list can be extended, but it will hardly be possible to integrate all the medical studies that would be relevant for this purpose.
  2. i) Is the research design appropriate? The research design follows the same pattern as performed in our previous investigation (see Madl et al., 2020).[1] Therefore we hope that the reviewer will accept it as it is. Otherwise, the entire experimental work needs to be redone and given the limited resources available at the moment this may put a complete halt to the research-work we are currently engaged in.
  3. i) Are the results clearly presented? In the rework process it became clear to us that the presentation of the results is in need of improvement. We have also improved this aspect in the current version.
  4. i) Are the conclusions supported by the results? Having eliminated redundancies in the previous manuscript, we hope that this section in the revised form is likewise better presented than before.
  5. i) Comments and Suggestions for Authors: The paper could be strengthened with addressing the points below:

Major:

  1. All the graphs are missing statistical analysis and number of repetitions.

The missing information have been added in the figure capture as well as in the graphs;

  1. Appendix A should be excluded. It is not based on any experimental results, and it is not too scientific. If want to include, experimental data should be presented:

The explanations of hormesis and QED given in the appendix refer to the focus of this special issue (The Emerging Role of Quantum Sciences and Radiation Biology in Biomedical Applications). However, in order to comply with the reviewer, we trimmed the appendix by skipping appendix B (will be used for a more theoretical paper that we are working on). The aspects currently included in the appendix highlight the relevance of our work from the perspective of quantum biology. We are of the opinion that without this additional information neither the hormetic principle nor QED can complement each other in a consistent way. The fact that the current paradigm in biology puts the emphasis preliminarily on the atomic/molecular aspect, whereas the field aspects that has been assigned its proper place in physics, has not yet seeped into biological sciences, deprives biological sciences of the full potential it may acquire when looking at both “sides of the medal”.

  1. It would be good to show results from HaCat cells + EM 

The results visible in figure 3 & 4 have been extended to include now also table 3 & 4 but unfortunately, they do not contain the requested data set. We apologize for that but during our lab work this particular aspect did pop into our minds. Nonetheless we appreciate this suggestion and will address that in future investigations. 

Minor:

  1. The figures are not in the right order. Starts from figure 2. 

Yes, we have become aware of that, please accept our apologies; the correct numbering of figures and tables has been implemented in the revised version.

  1. Please provide more details early in the text the electromagnetic set up and what is the field strength in mTesla of the EM. 

Thank you for this suggestion. We included these data into the abstract. Additional description has been provided in the methods section, the measured field intensities have been reworked on and are now accessible in table 6.

  1. Line 78 should be mentioned that the reduction was observed only after 48 hr and not 24 hours.

Thank you, has been corrected by introducing a revised figure 1 with the corresponding table.

  1. 4.2 still contain traces of the track changes in red and blue

We have removed all editing trackers in the current manuscript.

Finally, Figure 4 has also been reworked. Unfortunately, we made a mistake in the submitted version; i.e. instead of embedding the correct dataset in the graphing software package, an erroneous dataset was used that did not correspond to the laboratory results.  We apologize for this and have mapped the correct dataset in the current version.

Hoping that the editing done in the revised version of our manuscript will find the approval by the reviewer.

We looking forward to the reviewers reply

Kindest regards

Pierre Madl et al.

[1] Madl. P.; De Filippis, A.; Tedeschi, A. Effects of ultra-weak fractal electromagnetic signals on the aqueous phase in living systems: a test-case analysis of molecular rejuvenation markers in fibroblasts. Electromagn. Biol. Med., 2020, 39(3): 227-238.  doi: 10.1080/15368378.2020.1762634
